# CD8+ and Regulatory T cells Differentiate Tumor Immune Phenotypes and Predict Survival in Locally Advanced Head and Neck Cancer

**DOI:** 10.3390/cancers11091398

**Published:** 2019-09-19

**Authors:** Alessia Echarti, Markus Hecht, Maike Büttner-Herold, Marlen Haderlein, Arndt Hartmann, Rainer Fietkau, Luitpold Distel

**Affiliations:** 1Department of Radiation Oncology, Universitätsklinikum Erlangen, Friedrich-Alexander-Universität Erlangen-Nürnberg, D-91054 Erlangen, Germany; alessia.echarti@web.de (A.E.); markus.hecht@hotmail.com (M.H.); Marlen.Haderlein@uk-erlangen.de (M.H.); rainer.fietkau@uk-erlangen.de (R.F.); 2Department of Nephropathology, Institute of Pathology, Universitätsklinikum Erlangen, Friedrich-Alexander-Universität Erlangen-Nürnberg, D-91054 Erlangen, Germany; Maike.Buettner-Herold@uk-erlangen.de; 3Institute of Pathology, Universitätsklinikum Erlangen, Friedrich-Alexander-Universität Erlangen-Nürnberg, 91054 Erlangen, Germany; Arndt.Hartmann@uk-erlangen.de

**Keywords:** CD8+, regulatory T cells, FoxP3+, head and neck squamous cell carcinoma, immune dessert, inflamed, excluded

## Abstract

Background: The tumor immune status “inflamed”, “immune excluded”, and “desert” might serve as a predictive parameter. We studied these three cancer immune phenotypes while using a simple immunohistochemical algorithm. Methods: Pre-treatment tissue samples of 280 patients with locally advanced HNSCC treated with radiochemotherapy were analyzed. A double staining of CD8+ cytotoxic T cells (CTL) and FoxP3+ (Treg) was performed and the cell density was evaluated in the intraepithelial and stromal compartment of the tumor. Results: The classification of tumors as “immune desert” when stromal CTL were ≤ 50 cells/mm^2^, “inflamed” when intraepithelial CTL were > 500 cells/mm^2^, and as “excluded” when neither of these definitions met these cut off values allowed the best discrimination regarding overall survival. These groups had median OS periods of 37, 61, and 85 months, respectively. In “immune desert” and “immune excluded” tumors high Treg tended to worsen OS, but in “inflamed” tumors high Treg clearly improved OS. Conclusions: We propose that, in locally advanced HNSCC, the tumor immune state “inflamed”, “immune excluded”, and “immune desert” can be defined by intraepithelial and stromal CTL. Tregs can further subdivide these groups. The opposing effects of Tregs in the different groups might be the reason for the inconsistency of Tregs prognostic values published earlier.

## 1. Introduction

Immune checkpoint inhibitors have exceptional anti-tumor activity in numerous cancer types [1] and they play a central role in the quickly changing therapeutic landscape of medical oncology [2]. However, the response to checkpoint inhibition is limited to a subgroup of patients. In recurrent and/or metastatic head and neck squamous cell cancer (HNSCC) PD-1 inhibitors have response rates of around 13% in unselected patients [3] and also only 19% after PD-L1 based patient selection (CPS score ≥ 1) [4]. As this is not satisfactory, there is great need to identify novel biomarkers with predictive significance. Checkpoint inhibitors activate cytotoxic immune cells, including CD8+ T cells, which, accordingly, are potential predictors of tumor response [5].

A variety of immune markers exists that can be used to characterize the tumor immune status. Besides immune cells, cell surface structures, cytokines, and tumor genetics or the microbiome may be used [6]. In the context of immune checkpoint inhibitors, especially PD-L1, expressed on tumor cells or immune cells were used to try to predict the responsiveness. More detailed analyses in HNSCC patients focused on PD-L1 on CD8+ T cells [7]. In a recent review article, the variety of different markers was grouped to three main immunological tumor subgroups, namely “immune desert”, “immune excluded”, and “inflamed” cancers [6]. According to this classification, checkpoint inhibitors are especially suited for the therapy of “inflamed” tumors. As a very complex pattern of properties defines these subgroups, they are too complicated for response prediction in daily clinical routine. Therefore, simpler strategies are needed to differentiate tumor groups. A promising and simple method to differentiate groups could be the number of infiltrating CD8+ T lymphocytes (CTL), as one could speculate that low CTL counts might impair the effectivity of checkpoint inhibition. Accordingly, in a subgroup analysis of the Keynote-001 trial, it has been shown for non-small cell lung cancer that patients with tumors having high intratumoural CD8 infiltration benefit from PD-1 blockade [8]. The same has previously been described for melanoma that is treated with PD-1 blockade [5]. Strongly inflamed cancers might have particularly high antigenicity combined with a lack of effective immunity due to suppressive mechanisms, so that checkpoint inhibition might be particularly successful in these cases. Immunescape in tumors involves a variety of mechanisms, including the up-regulation of PD-L1, CTLA-4, TIM-3, LAG-3, IL-10, TGF-β and the presence of M2-macrophages as well as regulatory T cells (Treg), besides others. Treg are regularly observed at high frequencies in cancers [9,10,11,12]. Analysis of prognostic significance of FoxP3+ Tregs has led to highly variable results between different cancer types and, overall, a negative prognostic effect was associated with Tregs [13]. Thus, it appears to be sensible to consider immunosuppressive FoxP3+ regulatory T cells in addition to immune-stimulating CTL [13,14,15]. In HNSCC, there are contradictory results regarding the prognostic value of Treg [16]. In previous studies, we found no prognostic significance in low risk, high risk, and metastatic HNSCC cohort [17,18]. Others reported a positive correlation of high Treg counts with either favorable [19,20,21] or unfavorable prognosis [22,23,24]. To understand the reasons for such contradicting results would be of great value to clarify immunological intratumoural mechanisms. We sub-grouped a large cohort of patients with locally advanced HNSCC according to the number of infiltrating CD8+ cytotoxic T lymphocytes in different compartments and analyzed the outcome in the thereby defined subgroups in order to develop a simple algorithm to differentiate malignancies with different immune states. Numbers of Treg were used to assess possible prognostic differences in these subgroups.

## 2. Results

### 2.1. Distribution and Prognostic Value of CTL and Treg

The distribution of the CTL and Treg in the tumor was measured as cell density both in the stromal and intraepithelial compartment of the tumor (Figure 1A–C). CTL had a median density of 306.5 cells mm^−2^ in the stromal compartment and 235.5 cells mm^−2^ in the intraepithelial compartment. According values for FoxP3+ cells were 145.1 and 52.7 cells mm^−2^, respectively (Figure 1D). CTL were 1.3 times more frequent in the stromal than in the intraepithelial compartment (Figure 1E). FoxP3+ cells were 3.1 times more frequent in the stromal than the intraepithelial compartment. The CTL cells were 4.6 times and 2.0 times more frequent than FoxP3+ cells in the intraepithelial and stromal compartment, respectively (Figure 1F).

For both markers, the median rate of lymphocytes per mm^2^ was used as cut off to separate two groups. The overall survival curve for patients with high CTL counts lay above the curve for patients with low CTL counts both with regard to the epithelial (*p* = 0.184) and the stromal compartment (*p* = 0.101), but it did not reach statistical significance (Figure 1G,H). In the stromal compartment, high FoxP3+ rates were clearly linked with improved overall survival (*p* = 0.001) (Figure 1J), whereas in the intraepithelial compartment, FoxP3+ cells had no influence on outcome (*p* = 0.694) (Figure 1K).

### 2.2. Prognostic Value of “Immune Desert”, “Immune Excluded” and “Inflamed Tumors”

We subsequently defined three subgroups to stratify the tumors better for overall survival analyses. Tumors were divided into three subgroups, namely “immune desert”, “immune excluded”, and “inflamed” cases in order to recapitulate a proposed model of different immunological backgrounds in cancer [6]. We defined two cut off points, including less or equal 50 CTL per mm² in the stromal compartment and more than 500 CTL per mm² in the intraepithelial compartment, as mentioned above. Cases with less or equal 50 CTL in the stroma were included in the “immune desert” group, those with over 500 intraepithelial CTLs in the “inflamed” group. All of the cases meeting neither of the two definitions were included in the “immune excluded” group. Patients meeting “immune desert” criteria comprised the smallest group (*n* = 53/18.9%) and had an unfavorable prognosis with a median survival of 37.0 month. The “immune excluded” group was the largest (*n* = 150/53.6%) and it had an intermediate survival of 61 month. The “inflamed” group (*n* = 77/27.5%) tended to have favorable overall survival of 85 month (*p* = 0.054) (Figure 2A). Cases in the unfavorable “immune desert” group presented with larger tumor sizes (*p* < 0.001) as compared to the other two groups (Figure 2B) and fewer p16 positive cancers when compared to the “inflamed” tumors (*p* = 0.007) (Figure 2C). There tended to be higher tumor grades in the “inflamed” group, which did not reach statistical significance (*p* = 0.121) (Figure 2D). No differences between the groups were observed regarding sex, age, location, presence of affected regional lymph nodes or distant metastasis (Appendix A). Cancer treatment differed between the groups with higher rates of definite radio-chemotherapeutic treatment in patients in the “desert” group (Appendix A).

In addition, we used the median value of FoxP3+ cells per mm^2^ to further subdivide the three earlier defined groups, in order to better understand the significance of FoxP3+ Treg. Tregs were separated into high and low at the median of each group. In the “desert” group numbers of Tregs had no influence on prognosis (*p* = 0.559) in the stromal compartment. In the intraepithelial compartment, the overall survival curve of patients with high Treg counts lay below the curve of patients with low Treg counts, without reaching statistical significance (*p* = 0.174) (Figure 2E,F). In the “immune excluded” group, low amounts of Tregs in the stromal compartment (*p* = 0.104) and high amounts of Treg in the intraepithelial compartment seemed to be unfavorable (*p* = 0.100), both not reaching statistical significance (Figure 2G,H). In the “inflamed” group, high amounts of Tregs were associated with improved survival both in the stromal and in the intraepithelial compartment, (*p* = 0.066 and *p* = 0.008, respectively).

### 2.3. Characteristics of “Immune Desert”, “Immune Excluded”, and “Inflamed Tumors”

Next, we studied the relative distribution of Tregs and CTL in the three groups defined by CTL infiltration. In the “desert” group, which is defined by low stromal CTL counts, both intraepithelial and stromal Tregs and CTL counts were low (Figure 3A). In the “immune excluded” group, stromal CTL and Tregs were increased to an equal extent, whereas in the intraepithelial compartment, both cell types were still quite low (Figure 3B). In the “inflamed” group, defined by high intraepithelial CTL, both in the intraepithelial and in the stromal compartment, CTL and Tregs were increased (Figure 3C). with the proportion of CTL being even higher than of Treg, in both compartments.

Tumor and epidemiologic characteristics of the groups defined by CTL and Treg infiltration were subsequently analyzed. In the “inflamed” group, low stromal Treg counts were associated with larger tumor size (*p* < 0.001) and more lymph node metastases (*p* = 0.039) (Figure 3D,E). Additionally, higher Treg counts in the “desert” group were linked to more undifferentiated cancers, as indicated by the tumor grading (*p* = 0.018) (Figure 3F). Some slight differences were observed with regard to the remaining parameters (Appendix A) without reaching significance.

## 3. Discussion

We studied the tumor samples of 280 head and neck cancer patients double stained for CD8+ and FoxP3+ T lymphocytes. First, we used the median as cut-off to separate patient groups with low and high numbers of lymphocytes quantified both in the epithelial and stromal compartment separately. As expected, in the stromal as well as intraepithelial compartment high numbers of CTLs were associated with improved OS. This confirmed the findings of several earlier studies [17,25,26]. CTLs are able to kill infected, neoplastic, or otherwise targeted cells by direct lysis, and thereby are the most powerful anticancer mediators of the immune system [27]. However, CTLs can only effectively destroy tumor cells if they can proliferate, mature, and are not inhibited in their functions by other factors. In this process, Tregs are one of the key players. They act as suppressors of inflammatory and anti-tumoral response, and thus maintain immunological tolerance to host tissues [13] and can potentially promote tumor growth. Thus it is not surprising that Tregs were often associated with an unfavorable outcome in malignancies, including HNSCC [24,28]. However, other studies exist that, in contrary, found an association of high numbers of Tregs and a favorable outcome [9,19]. This positive impact of FoxP3+ Tregs might be partially attributed to the down-regulation of a persistent inflammatory process, which could promote inflammation-driven tumor progression [13]. In the present study, high numbers of Tregs in the stromal compartment were clearly associated with an improved OS (*p* = 0.001). Astonishingly, intraepithelial Tregs had no impact on OS (*p* = 0.694), going in line with earlier findings in a rectal cancer cohort with 103 patients [29]. So far, it remained unclear why such powerful immune suppressors as FoxP3+ Tregs, which are located in the critical intraepithelial area, have no impact on survival. It has to be mentioned that Tregs were solely detected by the positivity of the FoxP3 staining and no additional markers as CD4 and CD25 were used. These three markers are typically used in flow cytometric assays [30]. However, FoxP3 appears to be the best single marker for Treg [31].

To figure out why there are such contrary effects, we decided to focus on the distribution of CTLs. Cancers were classified in only two groups by low or high amounts of tumor infiltrating inflammatory cells may not appropriately address the problem. Recently, three groups of inflammatory states in tumor microenvironment were described, defined by the frequency and properties of infiltrating cells [6]. We used an approach to appropriately define these groups by the frequency of CD8+ cells as a surrogate marker. These groups were called “immune desert”, with nearly no immune cell infiltration, “immune excluded” in which the immune cells are concentrated at the stromal-epithelial boundaries with low intraepithelial counts and “inflamed”, in which the immune cells infiltrate the epithelial compartment and strong inflammation is present. Here, we used more than 500 intraepithelial CTLs mm^−2^ to define the “inflamed” group, less or equal 50 CTL stromal cells mm^−2^ to define the “desert” group and included all cases not meeting one of the above-mentioned criteria in the “immune excluded” group. These thresholds for the three groups were fitted for the present cohort and they must be verified in future studies. However, it demonstrates that the three groups can be differentiated that have different prognostic parameters.

The survival plots of the “immune desert”, “immune excluded”, and “inflamed” group were further subdivided according to the numbers of infiltrating FoxP3+ Tregs. The most striking finding was that intraepithelial Tregs gained prognostic relevance when analyzing the subgroups in contrast to the whole cohort and that the prognostic value of Tregs is largely dependent on the state of inflammation of the tumor. In the “immune desert” and “immune excluded” group, higher intraepithelial Tregs than the median tended to worsen overall survival. This is in accordance with the classical assumption that immunosuppressive Treg are associated with a worse prognosis of patients by suppressing anti-tumoural immunity [32]. In contrast, in “inflamed” tumors, high Tregs were associated with significantly better overall survival, probably due to the down-regulation of harmful inflammatory reaction, which could favor tumor progression, as inflammation can induce tumorigenesis [9].

As the unfavorable effect in the “desert” and “excluded” groups and the favorable effect in the “inflamed” group opposed each other in the undivided group, those effects were obscured when analyzing the complete cohort divided at the median of CTL counts. This finding might be the key to explaining the differing findings of the prognostic value of Treg in different published analyses mentioned above. Our cohort consists of different primary tumor sites, stages, and includes different therapy regimens that result in a cohort with different states of inflammation. If uniform groups are studied, the states of inflammation might be more consistent and result in either favorable or unfavorable prognoses.

Especially in “immune excluded” tumors, the prognostic effect of Tregs differed when comparing the stromal and intraepithelial compartment. High Treg counts improved prognosis in the stromal compartment, but worsened it in the intraepithelial compartment. The reason might be that the accumulation of immune cells plays different mechanistic roles in the stromal and intraepithelial compartment of the tumor. In the stroma, both endothelia and fibroblasts and their surface markers are important in the recruitment of inflammatory cells [6,33]. Moreover, the extracellular matrix, especially collagens and fibronectin, are playing an essential role. The main difference to intraepithelial inflammation is probably that no direct interaction of tumor cells and inflammatory cells takes place and the chemokine concentrations are lower. In intraepithelial recruitment, the tumor cell chemokines that are produced by tumor cells are higher concentrated and direct cell-to-cell, receptor mediated interactions are possible. Thus, it can be speculated that intratumoural CTL more frequently have anti-tumoural properties than stromal CTL. This could explain why Tregs inhibiting intratumoural (mostly anti-tumoural directed) CTL worsen prognosis and Tregs inhibiting stromal CTL (unspecific inflammation) improve prognosis by the reduction of growth factors and cytokines [9].

The results of simultaneous FoxP3-staining indicate that the very simple classification that is based on CTLs alone might not suffice to define groups predicting response to cancer therapies. Analysis with regard to the numbers of Tregs indicated that both the “excluded” and the “inflamed” groups were not homogenous, but that they can be further stratified according to FoxP3+ Treg cell counts, underlining an important immune modulatory role of Tregs in the tumor microenvironment. Hence, one could postulate that in the different subgroups the success of checkpoint inhibition might be influenced by the amounts of Tregs. Accordingly, murine models resistance to PD-1/PD-L1 pathway blockade is frequently mediated by Treg [34,35]. Thus, inflamed tumors with low intraepithelial Tregs might probably respond best to the immune checkpoint blockade. However it is possible that checkpoint inhibitors can restore CTL activity, even in the “immune desert” and “immune excluded” group with low Treg.

Plentiful data exists reporting improved treatment response to immune checkpoint inhibitors in inflamed tumors when compared to not-inflamed tumors [6]. Our findings indicate that it is probably insufficient to select cancer patients for checkpoint inhibitor therapy by CTL counts or degree of inflammation alone. A variety of mechanisms are probably determining tumor response to checkpoint inhibition due to the complexity of the interplay of immunocytotoxicity and immunosuppression. The subgrouping by Treg possibly improves the selection of patients that will benefit from checkpoint inhibition, but still will not represent the whole story. Prospective studies using checkpoint inhibitors are needed to develop an immunohistochemical algorithm beyond PD-L1 stainings to select possible responders.

## 4. Materials and Methods

### 4.1. Patient Cohort

Tumor tissues of 280 patients that suffer from locally advanced HNSCC treated at the University Hospital Erlangen between 1992 and 2010 were collected [17,18,36]. Clinical data were obtained from the Erlangen Tumour Centre Database and patient records. Written informed consent was obtained ‘front door’ from all patients allowing for the collection of their tissue and clinical data. The use of formalin fixed paraffin-embedded material from the Archive of the Institute of Pathology was approved by the Ethics Committee of the Friedrich-Alexander-University of Erlangen-Nuremberg on 24 January 2005, waiving the need for consent for using the existing archived material. Table 1 gives clinical and histological characteristics of the patient cohort. 51 patients received a definitive radiochemotherapy (RCT), 196 were treated with surgery and adjuvant RCT, and 33 with a neoadjuvant RCT followed by surgery. In the subgroup that was treated with adjuvant RCT, 173 patients (88.3%) had no residual tumor, 21 patients (10.7%) had microscopic, and 2 (1.0%) macroscopic residual tumor. All of the patients were in first line treatment and all tissue samples were obtained prior to RCT. The median follow-up was 44 months. Most of the patients were male (82.9%) with a median age of 55 years. All T-stages were common; most patients had N2 lymph node status (58.6%). Overall survival was 51.2% after five years. The distant metastases free rate was 72.4% and the local recurrence free rate was 76.3% after five years (Figure 1A).

### 4.2. Immunohistochemistry

Formalin-fixed paraffin-embedded tissue specimens were processed in tissue microarrays with cores of 2 mm diameter (Figure 1B). CD8+ antibodies were used to identify cytotoxic T lymphocytes (CTL, blue membranous staining) and FoxP3 antibodies to identify regulatory T cells (Treg, red nuclear staining). Immunohistochemistry was performed as double staining (Figure 1C) while using a CD8-specific antibody (M7103, Agilent, Santa Clara, CA, USA) and a FoxP3-specific antibody (Ab20034, abcam, Cambridge, United Kingdom). For detection, a Polymer-Kit and Fast Red and Polymer-Kit and Fast Blue (POLAP-100 Zytomed Systems, Berlin, Germany) were used according to the manufacturer’s instructions [37,38]. A microscopic scanner (Imager Z2, Zeiss, Göttingen, Germany), combined with a Metafer software (Metasystems, Altlussheim, Germany), was used to scan the stained TMAs at a magnification of 400 times. The cells were counted applying the image analysis software (Biomas Software, Version 3.3; MSAB, Erlangen, Germany) [39]. The stromal and the epithelial compartment were both analyzed separately and positive cells per mm^2^ were quantified.

### 4.3. Determining the Cut of Values for “Immune Desert”, “Immune Excluded” and “Inflamed Tumors”

In a recent review article, three main immunological phenotypes of cancer have been described: “immune desert”, “immune excluded”, and “inflamed” [6]. The aim of the current analysis was to classify tumors in these three groups that were based on intratumoral CTL and Treg. For determining the cut of values, first we decided that “immune desert” tumors should be defined by stromal and “inflamed” tumors by intraepithelial CTL. Randomly, the “immune desert” group was defined as < 10 CTL mm^−2^ stroma and the “inflamed” group as > 1000 CTL mm^−2^ in the epithelium and the “immune excluded” group as not meeting either parameter. Kaplan Meier plots for overall survival were used to find a possible difference between the “immune desert” and “immune excluded” by changing the threshold in steps of 10 CTL. After finding a clear difference (*p* < 0.010), we subsequently repeated this approach comparing “immune excluded” and “inflamed” groups while using steps of 50 CTL. The cut of values that were found by this procedure were used in a second round repeating the same procedure, which resulted in cut of values of ≤ 50 cells mm^−2^ in the stromal and > 500 cells mm^−2^ in the epithelial compartment as the best discriminative values regarding overall survival.

### 4.4. Statistical Analysis

Statistical analysis was performed while using SPSS version 21 (IBM Inc., Chicago, IL, USA). Overall survival (OS) was analyzed with the Kaplan Meier method. The log-rank test was used to compare the survival curves between the subgroups of the patient cohort. The comparison of the characteristics of the different immunological groups was done with the Student’s *t*-test.

## 5. Conclusions

In conclusion, the present head and neck cancer cohort can be divided in at least three different subgroups, which might overlap with the earlier described phenotypes of “immune desert”, “immune excluded”, and “inflamed” tumors. Useful cutoff values may be ≤ 50 stromal and > 500 intraepithelial CTLs mm^−2^. These cut off values must be validated in future studies. Treg counts can further subdivide these groups and indicate that there is still heterogeneity in the groups. The different prognostic findings for Tregs in the subdivided groups might explain the contradictory results of previously published studies regarding the prognostic significance of Treg.

## Figures and Tables

**Figure 1 cancers-11-01398-f001:**
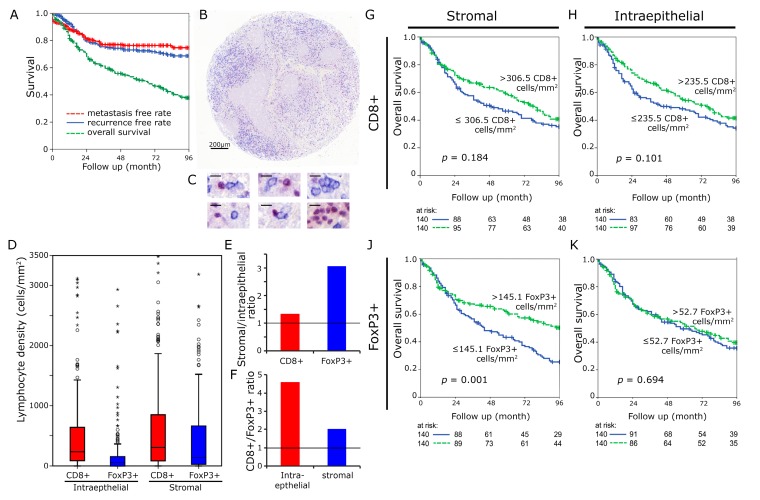
CD8+ and FoxP3+ cell densities in head and neck cancer. Kaplan Meier plots for metastasis free rate, recurrence free rate and overall survival in the complete cohort of 280 patients (**A**). Tissue samples were processed into tissue microarrays using a core diameter of 2 mm. Scale bar 200 µm (**B**). High power views (1:400) with immunohistochemical double staining for FoxP3 (red nucleic staining) and CD8 (blue predominantly membranous staining). (Scale bars 10 µm) (**C**). Lymphocyte densities (cells/mm²) in the intraepithelial and stromal compartment were separately analyzed (**D**). Stromal/intraepithelial ratio of CD8+ and FoxP3+ cells (**E**). CD8+/FoxP3+ ratio in intraepithelial and stromal compartment (**F**). Kaplan Meier plots for densities of CD8+ lymphocytes in the stromal (**G**) and intraepithelial H as well as FoxP3+ cells in the stromal (**J**) and epithelial (**K**) compartment (overall survival). Cut-off values were the median.

**Figure 2 cancers-11-01398-f002:**
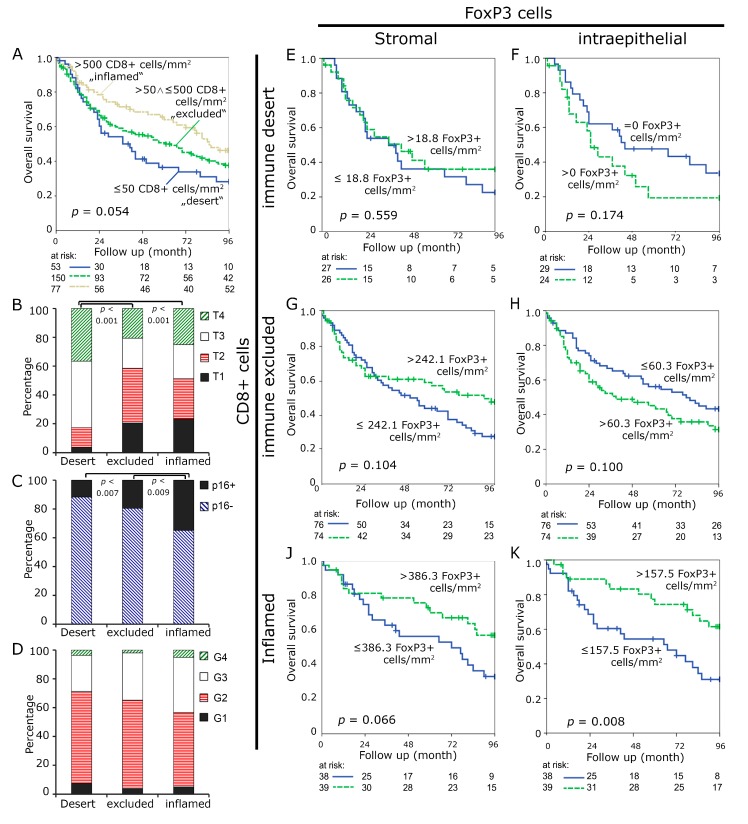
Subgroups “immune desert”, “immune excluded” and “inflamed”. “Immune desert” was defined as having ≤ 50 CD8+ cells/mm^2^ in the stromal compartment, “immune excluded” ≤ 500 CD8+ cells in the intraepithelial and > 50 in the stromal compartment and “inflamed” having > 500 cells in the intraepithelial compartment. Kaplan Meier plots for overall survival in these groups (**A**) Tumor size (**B**) p16 staining (**C**) and tumor grade (**D**) were compared. Kaplan-Meier plots for the FoxP3+ cell densities in the stromal and intraepithelial compartment in the “immune desert” group (**E**,**F**), in the “immune excluded” group (**G**,**H**), and the “inflamed” group (**J**,**K**) Cut-off values were median FoxP3+ densities.

**Figure 3 cancers-11-01398-f003:**
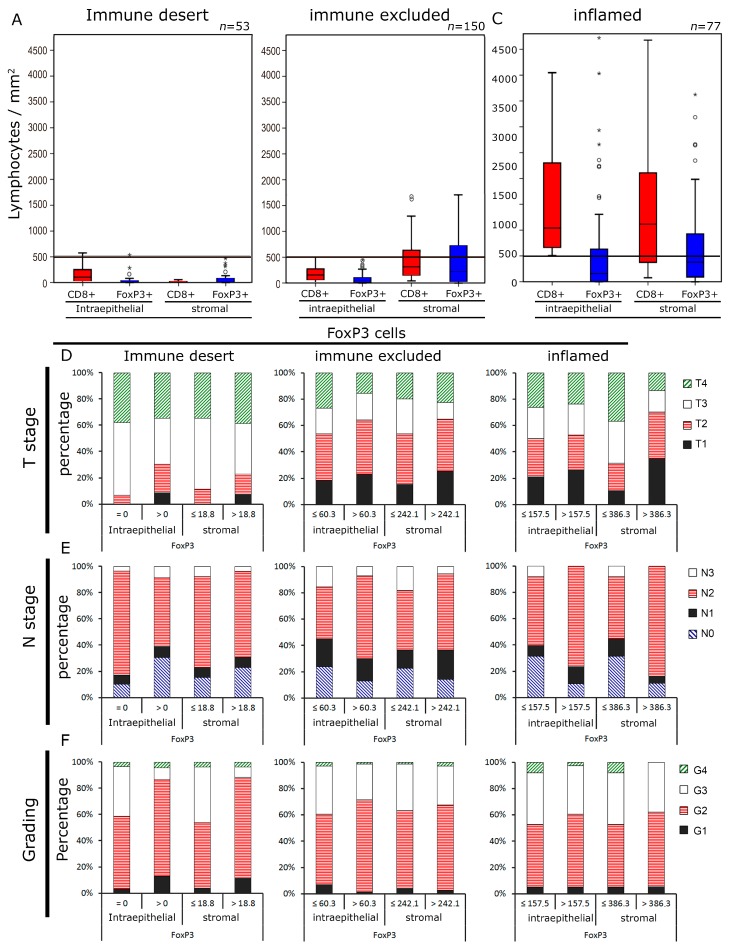
Lymphocyte densities and clinical characteristics in the “immune desert”, “immune excluded” and “inflamed” groups. Lymphocyte densities (cells mm^−2^) in the intraepithelial and stromal compartment of the “immune desert”. (**A**) the “immune excluded”. (**B**) and the “inflamed” group. (**C**) Tumor size. (**D**) N stage E and tumor grade. (**F**) in the three subgroups depending on the FoxP3+ densities are depicted. Cut-off values were the median FoxP3+ densities.

**Table 1 cancers-11-01398-t001:** Clinical characteristics of 280 head and neck cancer patients.

Clinical Characteristics	Number of Patients (Percentage)
Sex	Male: 232 (82.9%), female: 48 (17.1%)
Age (years)	mean: 55 ± 9, min.: 27 max.: 81
Primary Tumor	T1: 52 (18.6%), T2: 85 (30.4%), T3: 73 (26.1%), T4: 70 (25%)
Regional Lymph Nodes	N0: 53 (18.9%), N1: 40 (14.3%), N2: 164 (58.6%), N3: 23 (8.2%)
Distant Metastasis	M0: 276 (98.6%), M1: 4 (1.4%)
UICC Stage	I 10 (3.6%), II 26 (9.3%), III 51 (18.2%), IV 193 (68.9%)
Grading	G1: 15 (5.4%), G2: 165 (58.9%), G3: 91 (32.5%) G4: 9 (3.2%)
p16	negative: 170 (60.7%), positive: 49 (17.5%), unknown: 61 (21.8%)
Treatment	definitive RCT: 51 (18.2%), adjuvant RCT: 196 (70%), neoadjuvant RCT: 33 (11.8%)
R classification (Adjuvant RCT)	R0: 173 (88.3%) R1: 21 (10.7%) R2: 2 (1.0%)

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
