# Peer review of "CD8+ and Regulatory T cells Differentiate Tumor Immune Phenotypes and Predict Survival in Locally Advanced Head and Neck Cancer"

_cancers, 2019, doi:10.3390/cancers11091398_

Round 1
Reviewer 1 Report
The authors aim to establish a new prognostic marker system for HNSCC by measuring the immune infiltrate. This approach is interesting and justified in times of immunotherapy. They describe three main groups, which they define as "immune desert", "immune excluded" and "inflamed". In addition, they have measured the influence of Treg in these groups and present interesting insights into the open question of whether Treg improve or worsen the prognosis.
Although I found the manuscript interesting and understandable in many parts, I have several concerns, which I will discuss in the following.
The manuscript has to be rewritten by a native English speaker. For example, the first sentence of the results section in the abstract is almost impossible to understand. The graphical abstract adds no further understanding and the graphic is sloppy. Why did the authors decide to devide the immune infiltrate in “immune desert”, “immune excluded” and “inflamed”? There have been published several classifications and the authors should discuss these and justify their decision. In the introduction from line 50 on the authors speculate on CD8+ T cells being the best predictive marker for a successful checkpoint inhibition. The section lacks solid literature review. And, with respect to the CheckRad CD8 trial, I am sure that the authors know all the details on this topic. It is not clear why Treg were defined as FoxP3+ as there have been published other definitions. This should be discussed, at least. Further, one could think that Treg were CD8+ FoxP3+ as the methods mention a double staining. If possible, I would recommend CD4 staning. Statistics are a big problem. Authors should seek statistical advice. They speak of trend and tendency for p-values not even near 5%. Generally spoken, there is no such thing as a tendency in statistics. Test turn out either significant or not. Less significant values do not depreciate the value of the paper! The term “associated” implicates significance. This has to be corrected in the respective sentences. T-test is only fitting for normally distributed variables. In Figure 3, the authors should adjust the scales of A, B and C to the same values. This would make it easier to understand. In the discussion line 161, there is a mental leap to Treg. Connect the parts. In line 214 the authors speculate on the role of an inflamed phenotype being target for a successful checkpoint inhibition. This has been published before and should be discussed. The R-Status of the patients treated with surgery should be stated. In the results one would think that the definition of the immune phenotype has been published, however in the methods this is described as established by the authors. This should be clarified. If the authors aim to establish a prognostic score, it might be helpful to divide the cohort in a screening and a validation cohort. In a random sample I found wrongly cited references. This has to be carefully corrected.
Reviewer 2 Report
In their paper, Echarti and colleagues address immune cell infiltration into locally-advanced HNSCC tumors, proposing a stratification strategy based on the CD8 and FoxP3 markers. Their main conclusion is that the degree of immune cell infiltration predicts survival and, potentially, treatment response, which is an interesting and clinically relevant finding. The authors have used state-of-the technology and the scientific and statistical methods appear to be sound. From my side, the paper could be a valuable contribution to Cancers provided that the following concerns are adequately addressed:
-the title is hardly appealing and also does not transmit a clear message. It needs to be improved.
-the authors could do a better job in highlighting the novelty of their findings: For example, it is well-known that the response to cancer treatment (particularly immunotherapy) is dependent at least in part on the degree of immune cell infiltration prior to treatment. What is there unique selling point?
-the authors begin the Introduction section with thoughts on immune checkpoint inhibitor therapy but their data are not based on checkpoint inhibitor-treated patients, so this kind of storytelling is questionable…
-the authors should discuss in more detail why the effects of Treg differ depending on the particular intratumoral localization (stroma vs. epithelium)
-the authors should more comprehensively discuss the determinants of immune cell infiltration in HNSCC (neoantigen load, tumor vasculature etc.). In doing so, they could also discuss papers such as Cheng et al., 2018, JACI (PMID 29391257), i.e., the contribution of stromal fibroblasts to immune cell infiltration and anticancer immunity.
Generally:
-is there a particular reason that authors do not show data on disease-free survival?
-line 36 – it should better read “quickly changing therapeutic landscape of medical oncology”.
-line 43 – I suggest to revise as follows “great need to identify novel biomarkes with predictive significance”.
-line 46 – change to “classified based on to the degree of immune cell infiltration”
-line 48 – “only suited” is debatable. I suggest to write “primarily suited”.
-line 60 – this sentence may be misleading because as written now, the reader could conclude that CTLs are immunosuppressive as well. This issue should be clarified.
-line 61 – the sentence “In HNSCC there are conflicting results” doesn’t tell anything at all. The sentence needs to be omitted or, alternatively, the “conflicting results” mystically referred to need to be specified.
-line 64 – the sentence starting with “Though it would be interesting…” is clearly not proper English and needs to be revised.
-line 109 – “worse tumor grades” is not an appropriate medical-scientific term. Tumor grades are “lower” or “higher” (not better or worse) – so change “worse” to “higher”…
-line 111 – change “gender” to “sex” (gender more refers to the perceived identity while sex specifically refers to the biological anatomy)
-line 123 – “ominous” is not an appropriate scientific term and is also not consistent with the data in Figure 2 J+K where a high Treg density associates with improved survival. The authors have to clarify this issue.
